

# Integrable Floquet dynamics

**Vladimir Gritsev[1][*] and Anatoli Polkovnikov[2]**

**1** Institute for Theoretical Physics, Universiteit van Amsterdam, Science Park 904,
Postbus 94485, 1098 XH Amsterdam, The Netherlands
**2** Physics Department, Boston University, 590 Commonwealth Ave., Boston, MA, 02215 USA

\* v.gritsev@uva.nl

## Abstract

We discuss several classes of integrable Floquet systems, i.e. systems which do not exhibit chaotic behavior even under a time dependent perturbation. The first class is associated with finite-dimensional Lie groups and infinite-dimensional generalization thereof. The second class is related to the row transfer matrices of the 2D statistical mechanics models. The third class of models, called here "boost models", is constructed as a periodic interchange of two Hamiltonians - one is the integrable lattice model Hamiltonian, while the second is the boost operator. The latter for known cases coincides with the entanglement Hamiltonian and is closely related to the corner transfer matrix of the corresponding 2D statistical models. We present several explicit examples. As an interesting application of the boost models we discuss a possibility of generating periodically oscillating states with the period different from that of the driving field. In particular, one can realize an oscillating state by performing a static quench to a boost operator. We term this state a "Quantum Boost Clock". All analyzed setups can be readily realized experimentally, for example in cold atoms.

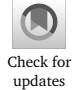

# 1  General Introduction

In classical systems integrability is a well defined concept [1]. A system with $2N$ degrees of freedom is called integrable if there exist $N$ independent functions, which have a vanishing Poisson bracket with the Hamiltonian. In this sense even trivial Floquet systems containing only two degrees of freedom like a Kapitza pendulum are nonintegrable because periodic driving eliminates energy conservation. Nevertheless stable *classical* Floquet systems are ubiquitous in nature and cover the whole range of scales from living species to planet and galactic systems. Absence of chaos in these systems guarantees long-term stability of structures around us. Classical Kolmogorov-Arnold-Moser theory and its ramifications provide rigorous tools for estimating required conditions for chaos to set up in perturbed *classically integrable* system. This theory shows that time-periodic perturbations only weakly affect the system thus preserving its stability for a long time. This stability can be associated with emergent approximate (Floquet) energy conservation law, where despite driving the stroboscopic dynamics of these systems is described by an effective (Floquet) Hamiltonian [2,3].

In quantum systems the very definition of integrability is not uniquely formulated [4]. In extensive translationally invariant systems an accepted definition of integrability is based on existence of local or quasi-local integrals of motion e.g. defining the generalized Gibbs ensemble [5], [6]. In general the notion of quantum integrability is defined only as an asymptotic statement typically either based on the classical or thermodynamic limit. Indeed an isolated finite dimensional quantum system like two spin 1/2 degrees of freedom can not be deemed integrable or nonintegrable because they form four discrete energy levels. It seems that in all known examples quantum integrability can be defined exactly in the same way as the classical integrability. Namely in the system with $N$ degrees of freedom one can require existence of $N$ independent functions of canonical operators (like coordinates and momenta or creation and annihilation operators), which commute with the Hamiltonian. In e.g. Ref. [7] it was shown how these operators can be explicitly constructed in a broad class of integrable systems. Another accepted route to defining quantum integrable systems, especially when integrals of motion are not a-priori known, is based on the Berry-Tabor conjecture (see Ref. [8] for the review). According to this conjecture generic integrable systems have Poisson energy level statistics as opposed to the nonintegrable systems, which have Wigner-Dyson random matrix statistics. While there is no proof that the two definitions of integrability are identical[1], practically the Berry-Tabor conjecture became a very powerful numerical tool in identifying quantum integrable systems.

---

[1]Moreover it is very easy to construct quantum systems with the Poisson level statistics, which nevertheless exhibit chaotic behavior in the classical limit.

Extending the notion of quantum integrability to time dependent, in particular Floquet, systems becomes even more tricky. One natural possibility is to require that there are integrals of motion commuting with the Floquet operator, i.e. the evolution operator within one driving period or equivalently commuting with the Floquet Hamiltonian [2,3]. Despite this seems like an obvious extension of normal equilibrium integrability, there is an important difference. Namely in this way one immediately looses connection with the classical integrable systems (see however [9], [10] for the counterexample when certain time-dependent integrable quantum systems can be connected to some classical integrable systems). Indeed with exception of trivial linear systems, like harmonic oscillators, the Floquet theorem for classical systems does not exist. Nevertheless one can construct examples of Floquet integrable systems in the thermodynamic limit like various driven band models or spin chains, which can be mapped to free particle systems [2, 11–13]. The second approach based on the Berry-Tabor conjecture is also possible to extend to Floquet systems by analyzing the statistics of the folded spectrum of the Floquet operator with the integrable and non-integrable regimes corresponding to the Poisson statistics and the statistics of the circular random matrix ensemble respectively [14–16]. Using this criterion any Floquet system, which can be mapped to a static system via a local rotation (e.g. a static system in the rotating frame) is integrable because its folded spectrum contains infinitely many level crossings [17, 18]. The Floquet integrable systems defined in this way do not heat up even at infinite times exhibiting localization in energy space [19–26], which in many respects very similar to the localization in real space in disordered models.

Defining integrability through the Poisson level statistics or absence of heating might seem universal, but there is a very important subtlety, which could make this definition different from traditional ones based on existence of local conserved operators (see e.g. discussion in Ref. [4] for non Floquet systems). The situation is very analogous to static integrable systems. For example, many body localized (MBL) systems can be regarded as integrable from the point of view of the level statistics and existence of quasi-local integrals of motion (see e.g. Ref. [27] for review). However, the integrals of motion in these systems can not be explicitly written as smooth analytic functions of the system size and other couplings unlike in integrable systems solvable by the Bethe ansatz. This is e.g. clear from the absence of the adiabatic limit (or equivalently absence of continuous transformations between eigenstates) in MBL systems, which immediately follows if we extend the arguments of Ref. [28] from Anderson insulators to MBL systems. Likewise the Kapitza pendulum, which has a stable non-heating regime at high driving frequencies, does not have differentiable smooth differentiable Floquet Hamiltonian and for this reason does not satisfy the adiabatic theorem [16, 18]. In particular, under infinitesimally slow adiabatic transformations the Kapitza pendulum will heat up to infinite temperature for any initial state [18]. Numerical studies confirm that same applies to interacting systems even in the parameter regimes, where the level statistics is perfectly described by the Poisson distribution [18]. So whether we discuss MBL systems or the Kapitza pendulum we are dealing with KAM type systems, which are stable against small integrability breaking perturbations in the statistical sense. Namely at given fixed parameters they have conserved local operators with the probability close to one but at the same time these conserved operators have infinitely dense set of non-analyticities everywhere. From these considerations it is very hard to formulate general conditions of integrability only based on the level statistics. Here we will rather focus on a narrower but well defined class of transitionally invariant Floquet integrable systems, where the integrals of motion can be found explicitly and which are smooth analytic functions of the parameters.

In this work we define and analyze in detail **three** generic classes of Floquet integrable systems in which one can define a local unfolded Floquet Hamiltonian. These systems by construction do not heat up and possess various properties shared with standard non-driven integrable systems. While these classes are definitely non-exhaustive (see e.g. Ref. [29]), they

provide a clear pass of constructing such systems. The first class is associated with the Hamiltonians, which can be represented as a linear combination of finite-dimensional Lie groups and their infinite-dimensional extensions. The other two, less obvious classes, are related to integrable statistical mechanical models. In particular, the second class corresponds to the Floquet operator realizing so called row transfer matrices and the third class, which we term"boost models", is associated with the corner transfer matrices. In boost models the Floquet Hamiltonian consists form the static part being a generic integrable Hamiltonian and time-dependent part being a boost operator, which in turn is closely related to the entanglement Hamiltonian. Although we focus on the Floquet systems, as it will become clear from our discussion, our results extend to more generic, e.g. non-periodic driving protocols. As a particular example we discuss emergence of oscillating state after a quench by a boost operator. We term this state as a "Quantum Boost Clock".

## 2 Floquet theory: setup

Let us briefly review some details about the Floquet theory and the high-frequency expansion, which will be important in the subsequent discussion (see Refs. [2,3] for more details). The Floquet theorem says that under the influence of a time-periodic Hamiltonian with period $T$, the evolution operator from the initial time 0 to some time $t$, $U(t,0)$, can be represented in the following form

$$U(t,0) = P(t)\exp(-itH_F),$$

where $P(t+T) = P(t)$ is a unitary periodic operator and the second term $\exp(-itH_F)$ effectively represents the time evolution with respect to the time independent (Floquet) Hamiltonian $H_F$. Note that in this form by construction $P(nT) = \mathbf{I}$, where $n$ is an integer and $\mathbf{I}$ is the identity operator. Therefore the stroboscopic evolution at discrete times equal to integer multiples of the period $T$ is described by the static Floquet Hamiltonian: $U(t = nT) = \exp(-itH_F)$. In this paper we will mostly analyze Floquet protocols corresponding to periodic step-like stroboscopic evolution. At the end we will comment on generalization of these results to more generic protocols. Note that integrability and absence of thermalization for the stroboscopic evolution automatically means integrability at arbitrary times in between simply because within one period the system does not have time to destroy conservation laws. At a more rigorous level the choice of the stroboscopic time is the gauge choice, which does not affect the spectrum of the Floquet operator [3].

For the step like drive between Hamiltonians $H_1$ of duration $T_1$ and $H_2$ for duration $T_2$ the Floquet Hamiltonian is defined as

$$\exp(-iH_F T) = \exp(-iH_1 T_1)\exp(-iH_2 T_2), \tag{1}$$

where $T = T_1 + T_2$. Generally, $[H_1, H_2] \neq 0$ which is the source of complexity. Here we try to identify those cases when the effective Floquet operator (and therefore the evolution operator) can be computed in a closed, yet possibly nontrivial form.

For our discussion of the effective Floquet Hamiltonian we will need one of the forms of

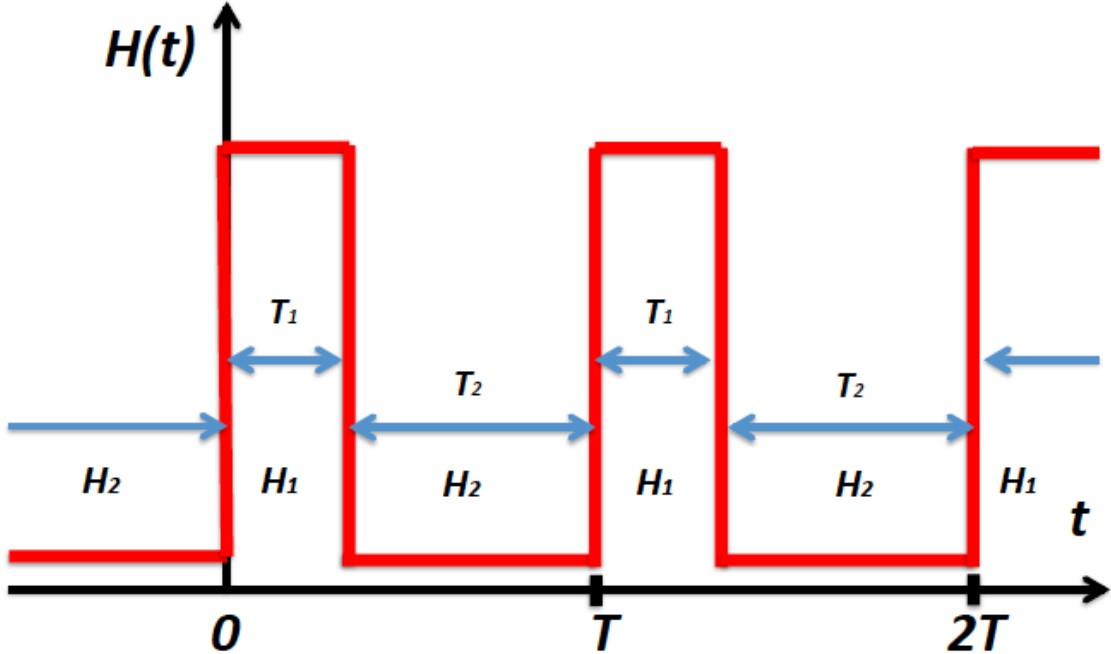

Figure 1: Periodic quench between non-commuting Hamiltonians $H_1$ and $H_2$ acting for durations $T_1$ and $T_2$ respectively. The whole system is time periodic with period $T = T_1 + T_2$.

the Baker-Campbell-Hausdorff (BCH) formula, namely

$$
\begin{aligned}
Z = \log(e^X e^Y) = {}& X + Y \qquad\qquad\qquad\qquad\qquad\qquad\qquad\qquad (2)\\
&+ \frac{1}{2}[X,Y] + \frac{1}{12}([X,[X,Y]] + [Y,[Y,X]])\\
&- \frac{1}{24}[Y,[X,[X,Y]]]\\
&- \frac{1}{720}([Y,[Y,[Y,[Y,X]]]] + [X,[X,[X,[X,Y]]]])\\
&+ \frac{1}{360}([X,[Y,[Y,[Y,X]]]] + [Y,[X,[X,[X,Y]]]])\\
&+ \frac{1}{120}([Y,[X,[Y,[X,Y]]]] + [X,[Y,[X,[Y,X]]]]) + \dots,
\end{aligned}
$$

where we identify $X \equiv -iH_1 T_1$, $Y = -iH_2 T_2$ and $Z = -iH_F T$. From this general formula it becomes clear that some internal structure of nested commutators must exist in order to be able to evaluate it in the closed form. Integrability of $H_F$ in this paper will be understood as existence of enough conserved integrals of motion to be able to diagonalize it.

In this paper we reveal several classes of integrable Floquet many-body quantum systems. The **first** class is formed by the models whose Hamiltonians are linear combinations of the generators of (in principle) arbitrary Lie algebras. For this class of Hamiltonians there is no distinction between quantum and classical dynamics, both of which map to a closed system of linear differential equations [31–35]. These generators can be always represented by the linear and bilinear forms of the creation-annihilation operators, for example using the bosons or fermions (see the second part of the book [31] and Ref. [36] for applications). The essential property of the Lie algebra formed by generators $\{J_k\}$ is existence of the bilinear product (commutator) which maps bilinear combinations to the linear one. One can generalize this by

considering the following structures for some operators $\{J_{k_j}\}$

$$[J_{k_1},[J_{k_2},\dots[J_{k_{n-1}},J_{k_n}]\dots]]=\sum_k c^k_{k_1,k_2,\dots,k_n}J_k, \tag{3}$$

where we have $n-1$ nested commutators on the left and $c^k_{k_1,\dots k_n}$ are the structure constants. In case when they vanish for certain $n$ the algebra is called nilpotent of order $n$. The case of $n=2$ defines the Lie algebra, while $n>2$ would define more complicated algebraic structures. For finite $n$ one can regard the operators that are coming out of $n-1$ commutators as additional elements of the algebra. Then if the Hamiltonian can be represented as a linear combination of these operators, the BCH expansion is going to produce some closed result. When $n=3$ one can find, for example, a realization of the algebra in terms of bosons $(b_p, b_p^\dagger)$ where $p=1,\dots,m$ and a Clifford algebra defined by the $r$-dimensional matrix representation $\Gamma^\mu$ and satisfying the relations $\{\Gamma^\mu,\Gamma^\nu\}=2\delta^{\mu\nu}$ where $\mu,\nu=1,\dots r$. Indeed, defining $J_\mu=\sum_{p,q=1}^m(\Gamma^\mu)_{pq}b_p^\dagger b_q$ one can show that they satisfy the following condition

$$[J_\mu,[J_\nu,J_\lambda]]=4J_\lambda\delta_{\mu\nu}-4J_\nu\delta_{\mu\lambda}, \tag{4}$$

We will not consider physical realizations of this mathematical structure here, which might be useful for some parafermion models. The Lie algebras can also be infinite-dimensional, like e.g. Kac-Moody, Virasoro or $\mathscr{W}_\infty$ algebras [37]. In this work we will briefly discuss only one particular representative of these infinite-dimensional families, namely the Onsager algebra realized in the case of $n=4$ and which is relevant for the transverse field Ising model.

The **second** class of the integrable models we consider here is realized by the non-commuting operators $V=\exp(\alpha X)$ and $W=\exp(\beta Y)$ for some $X$ and $Y$, such that they correspond to addition of rows of horizontal and vertical edges in *integrable* classical 2D (square) lattice models. By standard quantum-classical correspondence this class of Floquet systems can be identified with 1D quantum integrable lattice models after the analytic continuation of $\alpha=-iT_1$ and $\beta=-iT_2$ to the complex plane. In the Floquet language these models correspond to switching between the Hamiltonians realizing the transfer matrices (see Fig. 1).

In the theory of classical integrable lattice models two types of the transfer matrices are known: row-to row transfer matrices related to the second class and the corner transfer matrices. So, the **third** class of models we consider here is related to the corner transfer matrices and is defined in terms of the so-called *boost* operators. While it does not generate a closed BCH series, it generates new integrals of motion at every step of the BCH iteration. So in these systems the Floquet Hamiltonians can be represented as a weighted sum of the boost operator and all integrals of motion. We note on passing that for the lattice models we mention in this context the boost operator is equivalent to the entanglement Hamiltonian. It is interesting that the construction similar to the one which underlies our boost models has recently appeared in a totally different context of quenches [38]. There after a quench (of arbitrary non-integrable Hamiltonian) the wave function at all times can follow the ground state of a certain local time-dependent Hamiltonian. The latter is obtained by applying the BCH series to the original Hamiltonian[2].

We note that these classes likely do not exhaust all possible Floquet integrable systems. In particular we think that at least some of the known discrete integrable systems, discovered and studied in the past [39] could be related to some physically-relevant Floquet integrable systems. These classes of integrable protocols can be realized in different physical settings. This can be clearly visualized with the Lie-algebraic models and with the Ising-related models as we will discuss below. As such protocols avoid heating effects they can be very useful for digital quantum simulations [40] by allowing one to use relatively large Trotter steps.

---

[2]We were not aware of this work before our paper was essentially completed.

Finally let us note that as with any other driven systems the physics can strongly depend on initial conditions, which can be also integrable or non-integrable. The formalism of integrable boundaries in integrable models has been introduced by Sklyanin [41] for the lattice models and extended by Ghoshal and Zamolodchikov [42] for the field-theoretic integrable models (like e.g. the sine-Gordon model). In the context of Conformal Field Theory these states are called Ishibashi states, and their special linear combinations are the Cardy states and represent the subject of the Boundary Conformal Field Theory, see e.g. [43] In string theory they correspond to the D-branes. In the studies of quench dynamics of integrable models these states become *initial* states [44] - in this case the time evolution can be analyzed explicitly. In the context of integrable lattice Statistical Mechanics models these special states are also well known - in particular, the six-vertex model with the *domain wall* boundary condition is also integrable [45]. In this latter context these states correspond to the integrable initial states for our second class of integrable Floquet models discussed below. Nonintegrable initial conditions of course will not cause heating in integrable Floquet systems, but generally will make their time evolution analytically intractable.

The paper is organized as follows. First we discuss a class of Lie-algebraic models of Floquet dynamics and provide an *algebraic* classification of different Floquet systems which is complementary to existing topological (cohomological) classification therein. Next we discuss the second class of the models related to the row-to-row transfer matrices of classical integrable lattice models. Finally we introduce a third class of models, related to the corner transfer matrices.

# 3 Lie-algebraic integrable Floquet Hamiltonians

## 3.1 Finite-dimensional Lie-algebraic Hamiltonians: dynamical symmetry approach

One (almost obvious) class of physical models where the BCH expansion can be summed up and which allows for computing a time-ordered exponent is a class of Hamiltonians which can be represented as a linear combinations of generators of some finite-dimensional Lie algebra $g$,

$$H_{Lie} = \sum_{k=1}^{N} a_k(t) J_k, \quad [J_k, J_l] = f_{kl}^{p} J_p. \tag{5}$$

Here the structure constants $f_{kl}^{p}$ $(k, l, p = 1, \ldots N)$ define the $N$-dimensional Lie algebra (possibly non-compact). Physically relevant examples of Lie-algebraic Hamiltonians are spin in arbitrary time-dependent magnetic field, quadratic fermionic and bosonic models (with finite number of different modes) with time-dependent couplings. In particular, non-interacting topological insulators obviously belong to this class of models. Also obviously all driven finite-dimensional quantum systems belong to this class. E.g. any Hermitian $N \times N$ matrix can be spanned by the generators of $SU(N)$ group and the identity. In order to distinguish integrable and non-integrable finite systems one has to require that $N$ is much smaller than the size of the Hilbert space, which typically scales exponentially with the number of degrees of freedom (system size). Therefore if $N$ scales linearly (polynomially) with the system size we would still term the corresponding system integrable.

We can assume that the functions $a_k(t)$ are time periodic:

$$a_k(t + T) = a_k(t). \tag{6}$$

If the Hamiltonians $H_1$ and $H_2$ belong to the same algebra, the evolution exponents for infinitesimal time steps belong to the corresponding Lie group and so their arbitrary product is also some element of the group. This implies that time-ordered exponent for arbitrary time-dependent coefficients is *some* element of the group,

$$U(t) = \mathcal{T} \exp\left(-i \int_0^t d\tau H_{Lie}(\tau)\right) = \prod_{k=1}^N \exp(-i\alpha_k(t)J_k) = \exp\left(-i\sum_k \Phi_k(t)J_k\right). \qquad (7)$$

Here the set of functions $\alpha_k(t)$ and $\Phi_k$ are related to functions $a_k(t)$ via solution of the Maurer-Cartan differential equations [34]. They can be directly obtained by substituting the ansatz above to the Schrödinger equation:

$$iU^{-1}(t)\partial_t U(t) = H_{Lie}(t). \qquad (8)$$

In practice one can always compute the disentangling functions $\alpha_k(t)$ for the low-$N$ dimensional Lie groups [46], while the functions $\Phi_k(t)$ defining the Floquet Hamiltonian

$$H_F = \frac{1}{T}\sum_k \Phi_k(T)J_k$$

are rigorously speaking defined only close to the group identity. We note that the procedure of disentanglement of the time-ordered exponent can be traced back to Feynman, see [47].

The operator $U(t)$ realizes unitary representation of the group $G$ whose algebra is $g$. Elements of the group act by adjoint action on the elements of the algebra $g$,

$$U(g)J_kU(g^{-1}) = \sum_p S_{kp}(g)J_p \qquad (9)$$

for some function $S_{kp}(g)$, which depends on particular representation and group element $g$. By the right choice of $g$ and therefore of $S_{kp}(g)$, one can transform the Hamiltonian $H_F$ into some *special* form. This special form of $H$ is defined in such a way that the Hamiltonian can be written as a linear combination of generators (let's call them $R_p$), $\tilde{H} = \sum_{p=1}^k r_p R_p$ which commute with the Hamiltonian and among themselves. These commuting set of elements $R_p$, in the language of group theory, are called elements of the conjugacy classes of different *maximal Abelian subalgebras*[3]. The number $k$ here could be different from the algebra's rank. There are several classes of these maximal abelian subalgebras [48], [49]. A very important sub-class is given by the Cartan subalgebras. In case of the compact or complex groups all Cartan subalgebras (subgroups) are conjugate (that is can be connected by a transformation from the group), and so there is only one Cartan subalgebra. In this case the Hamiltonian can be uniquely diagonalized. It is, however, known to be not true in general [50]. In fact different classes of conjugate Cartan subgroups are related to different series of unitary representations. For the *real forms* of Lie algebras the number of different Cartan subalgebras is not unique. The number of conjugacy classes of Cartan subalgebras is finite for finite dimension $N$. Its characterization for classical algebras of $A - B - C - D$ (as well as for exceptional) types is given in [51], [52] and summarized in Appendix (see Table 1) (exceptional algebras are not included there). In practice it implies that depending on the form of coefficients $a_k(t)$ adjoint action of the group elements can bring $H$ to a different Cartan subalgebras. Therefore spectra of different Floquet systems are in correspondence with different Cartan subalgebras.

Second important class of maximal abelian subalgebras are nilpotent abelian subalgebras. They generate nilpotent orbits in the group $G$ which have rich topological properties and can be classified by partitions of $N$ for the complex and real cases [53].

---

[3]An Abelian subalgebra $m$ of a Lie algebra $g$ is a maximal Abelian subalgebra if it is equal to its own centralizer in $g$: $[m, m] = 0$, and $\text{cent}_g m = m$ where $\text{cent}_g m = \{x \in g | [x, y] = 0, \forall y \in m\}$.

## 3.2 Example: Mathieu harmonic oscillator

Let us illustrate these general statements on a simple, yet rich example. Following [54] we consider the harmonic oscillator $H = \frac{1}{2}(p^2 + \omega^2(t)x^2)$ with time-dependent frequency $\omega^2(t) = \omega_0^2(1 + h\cos\omega t)$, with $-1 < h < 1$. This problem can be solved by elementary methods but we will use it to illustrate the classification scheme. This Hamiltonian is a linear combination of the generators of the non-compact $SU(1,1)$ algebra spanned by three generators: $J_1 = \frac{1}{4}(p^2/\omega_0 - \omega_0 x^2)$, $J_2 = \frac{1}{4}(xp + px)$, $J_0 = \frac{1}{4}(p^2/\omega_0 + \omega_0 x^2)$. The Casimir operator is defined by the indefinite form $J_0^2 - J_1^2 - J_2^2$. As noted above, the solutions for the disentanglement functions $\alpha(t)$ can be obtained in terms of the solutions of the classical equation of motion (non-compact analogue of the Bloch equation), $\ddot{\xi}(t) + \omega^2(t)\xi(t) = 0$ coming from the Eq. (8), see Refs. [33] and [34] for more details. The evolution operator over a period $T$ (the monodromy matrix) according to Eq. (7) can be represented as $U(T) = \exp(-i\boldsymbol{\Phi} \cdot \mathbf{J})$, where $\boldsymbol{\Phi} \equiv \boldsymbol{\Phi}(T)$. The relationship between components of $\boldsymbol{\Phi} = \Phi\mathbf{n}$ and the two linearly independent solutions $(\xi_1(t), \xi_2(t))$ of the classical equation for $\xi(t) = C_1\xi_1(t) + C_2\xi_2(t)$, where $C_{1,2}$ are defined by the initial conditions (e.g. $U(0) = 1$) can be most easily obtained as follows. The form of the defining equations for $\xi$ and for $\mathbf{n}$ do not depend on representation. Therefore one can take the lowest possible one, defined by the Pauli matrices, $J_0 = \frac{\sigma_3}{2}, J_{1,2} = \frac{i\sigma_{1,2}}{2}$ and compare the "spinor" form for the evolution of $(\xi(t), \dot{\xi}(t))^T$ from the initial conditions. This leads to [54][4]

$$n_0 = \frac{1}{|\Delta|}(\dot{\xi}_1\dot{\xi}_2 + \xi_1\xi_2),$$

$$n_1 = \frac{1}{|\Delta|}(\dot{\xi}_1\dot{\xi}_2 - \xi_1\xi_2),$$

$$n_2 = \frac{1}{|\Delta|}(\xi_1\dot{\xi}_2 + \dot{\xi}_1\xi_2), \tag{10}$$

where $\Delta = \xi_1\dot{\xi}_2 - \dot{\xi}_1\xi_2$ is the Wronskian which is time independent, and the vector $\mathbf{n}$ is normalized as $\mathbf{n}^2 = -\text{sign}(\Delta^2)$ while $\Phi = 2T\epsilon\sqrt{\mathbf{n}^2}$. Here the Floquet eigenenergy $\epsilon$ is given by

$$\epsilon_\nu = \nu\frac{\Delta}{2T}\int_0^T \frac{dt}{|\xi|^2}, \tag{11}$$

where $\nu$ is a representation index defined as follows. The group $su(1,1)$ is non-compact - it is defined as group of transformations which preserve the bilinear form $\mathbf{n}^2 = n_0^2 - n_1^2 - n_2^2$ and therefore there are three cases depending on the sign of $\mathbf{n}^2$. Indeed, the non-compactness of $su(1,1)$ implies that the phase space is foliated into three geometrically different situations: the two-sheet hyperboloid, one-sheet hyperboloid and the cone (see e.g. [31], Chapter 5). This corresponds to three different situations for $\mathbf{n}^2$: positive, negative, or zero. The solutions of the equation for $\xi$ could be either complex (for stable region corresponding to $\mathbf{n}^2 = 1$) when however $\xi_1^* = \xi_2$ or real (for the regions of unstable motion, $\mathbf{n}^2 = -1$ or $\mathbf{n}^2 = 0$). The first case of positive $\mathbf{n}^2$ corresponds to the discrete series of the $su(1,1)$ representation. Then $\nu = \frac{1}{2}(n + \frac{1}{2})$, $n = 0, 1, 2, \dots$ and the Floquet Hamiltonian $H_F$ can be put to the form proportional to $J_0$ by the adjoint action of Eq. (9) which has, in this case, the form $U(g^{(1)}) = \exp(-i\alpha_0^{(1)}(t)J_0)\exp(-i\alpha_2^{(1)}(t)J_2)$. This is familiar bounded harmonic oscillator corresponding to the stable motion. In the second possible case $\mathbf{n}^2 = -1$ the Floquet spectrum $\epsilon_\nu$ is a set of continuous real numbers, $-\infty < \epsilon_\nu < \infty$. This corresponds to the unstable motion. The operator $\mathbf{n} \cdot \mathbf{J})$ can be transformed to the form proportional to the generator $J_1$. In

---

[4]Note small inconsistency in Eq. 38 of [54]. It is fixed here.

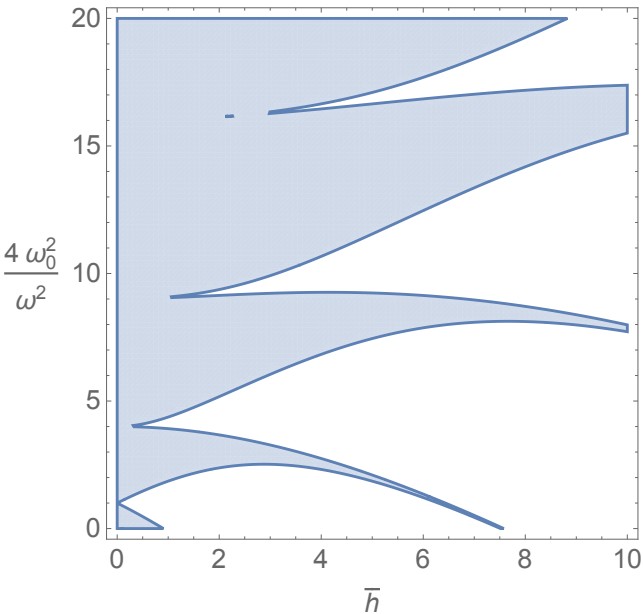

Figure 2: Stable (filled) and unstable (white) regions of parameter space of the Mathieu harmonic oscillator. Here $\bar{h} = -2\omega_0^2 h/\omega^2$. This picture represent three orbits of the $su(1,1)$ action: the stable region is the orbit of the $J_0$ Cartan subalgebra while the unstable one is the orbit of $J_2$ (which can also be represented by $J_1$. The boundary between stability and instability region correspond to the nilpotent subalgebra.

this case, the adjoint action $U(g^{(2)}) = \exp(-i\alpha_0^{(2)}(t)J_0)\exp(-i\alpha_2^{(2)}(t)J_2)$. Finally, in the third possible case, when $\mathbf{n}^2 = 0$ the operator $\mathbf{n} \cdot \mathbf{J}$ can be transformed to the form $J_0 + J_1 = p^2/2\omega$ by the adjoint action of $U(g^{(3)}) = \exp(-i\alpha_0^{(3)}(t)J_0)$. The operator $p^2/2\omega_0$ has a continuous spectrum $0 \leq \epsilon_\nu < \infty$ and corresponds to the boundary between the stable and unstable regions. The whole picture is illustrated on Fig. (2), where the regions of unstable motion (white areas) penetrate into the stable regions (filled). We note that this stability diagram has been recently confirmed experimentally in Ref. [55] with an extremely high accuracy. If we consider $2 \times 2$ matrix representation of the $su(1,1)$ algebra one can connect those *special forms* mentioned above with Cartan and nilpotent subalgebras. The domain of stable motion correspond to the Cartan subalgebra generated by $J_0$ while the unstable motion correspond to the Cartan subalgebra generated by $i\sigma$. The stability boundary correspond to the nilpotent subalgebra generated by $\sigma_+$. If we were to use the language of symplectic group we would have to multiply the Hamiltonian by the matrix $\mathscr{J}$ of the symplectic form $H \to \mathscr{J}H$. Then the stable region correspond to the *elliptic block*

$$\begin{pmatrix} \cos(2T\epsilon_n) & \sin(2T\epsilon_n) \\ -\sin(2T\epsilon_n) & \cos(2T\epsilon_n) \end{pmatrix}, \tag{12}$$

while the unstable region correspond to the *hyperbolic block* Finally, *parabolic blocks* are obtained by considering nilpotent subalgebras and correspond to the boundary between stable and unstable motions. For more information about Floquet analysis in terms of sumplectic block we refer to [56] (in particular see Sec. E. 2.2). Approach, similar to ours is developed in [48].

The main message of the example above is the following. Depending on the position in the phase diagram of Fig. (2) we can, by applying a suitable similarity transformation, diagonalize our Floquet Hamiltonian to one of the three types of the matrices considered above - elliptic, hyperbolic or parabolic. These matrices *can not* be continuously connected by the transforma-

tion from the $su(1,1)$ - they correspond to different *equivalence classes*. This can be visualized geometrically: the $su(1,1)$ consist of two regions (corresponding to stable and unstable motions) disconnected by the light cone (boundary situation). This example illustrates a general scheme for an arbitrary number of bosonic degrees of freedom when the Hamiltonian has a quadratic form. We note that the $su(1,1)$ is isomorphic to the $sp(2,R)$. For several (say $M$) bosonic degrees of freedom one can extend the above analysis of equivalence classes for the group $Sp(2M,R)$. This has been done in [50], [52] and summarized in the Appendix A (see also [48] for $M = 1, 2, 3$). According to the Table 1 there are $(M + 2)^2/4$ different Cartan subalgebras in the case of even $M$ and $(M + 1)(M + 3)/4$ in the case of $n$ odd. Our simplest example above has two non-equivalent Cartan subalgebras corresponding to the elliptic and hyperbolic blocks respectively.

## 3.3 Infinite-dimensional algebras

### 3.3.1 Self-dual systems and Onsager algebra

If the rank of the algebra is infinite and the algebra is not nilpotent, such that infinite number of commutators survive the BCH expansion (2) then in general the Floquet Hamiltonian can not be found in the closed form. However, there is a special class of infinite-dimensional algebras, where the integrability can still be established. This happens when the commutators have a certain recursive structure. In particular, this is the case for Onsager algebra [57] which is an underlying algebraic structure of the 2D Ising model. The "seeds" of the Onsager algebra is given by two operators

$$A_0 = \sum_{n=1}^{L} \sigma_n^x, \quad A_1 = \sum_{n=1}^{L} \sigma_n^z \sigma_{n+1}^z, \tag{13}$$

which generate an infinite dimensional algebra spanned by the basis $\{A_m, G_m\}$, $n \in \mathbb{Z}$

$$
\begin{aligned}
[A_l, A_m] &= 4G_{l-m}, \quad l \geq m \\
[G_l, J_m] &= 2J_{m+l} - 2J_{m-l}, \\
[G_l, G_m] &= 0.
\end{aligned}
\tag{14}
$$

Note that $G_{-m} = -G_m$.

In a remarkable paper [58] it was shown that the algebraic relations of the form

$$
\begin{aligned}
[A_0, [A_0, [A_0, A_1]]] &= 16[A_0, A_1], \\
[A_1, [A_1, [A_1, A_0]]] &= 16[A_1, A_0],
\end{aligned}
\tag{15}
$$

and those which follow from them, supply sufficient conditions for demonstrating integrability of the *self-dual* Hamiltonian $H = \alpha_0 A_0 + \gamma A_1$ (with real $\alpha$, $\beta$) such that there is a linear map $\tilde{\phantom{a}}$ (duality relation) which connects $A_0$ and $A_1$: $A_1 = \tilde{A}_0$ such that $A_0 = \tilde{A}_0$. Relations (15) and duality map provide sufficient condition to systematically construct an infinite number (for infinite system) of conserved charges $Q_m$. In particular, the transverse field Ising Hamiltonian

$$H = A_0 + \gamma A_1 \tag{16}$$

belongs to the family of conserved charges

$$Q_m = A_m + A_{-m} + \gamma(A_{m+1} + A_{1-m}). \tag{17}$$

Moreover in Ref. [59] it was shown that the relations (15) and the closure relations (e.g. specification of the boundary conditions) are enough to determine the form of the spectrum. We

note that all these constructions work for any type of the dual system (discrete or continuous) and in arbitrary number of dimensions.

There are two possible extensions of this self-dual model. One is related to the $sl(N)$ generalization of Onsager algebra [60] (a traditional Onsager algebra discussed above is related to the $sl(2)$ loop algebra, see [57]). In this case one can construct spatially inhomogeneous Floquet models of higher symmetry. A second extension is related to the $Z_N$ generalization of the Ising model (Ising model is a $Z_2$ model), a so-called chiral Potts model [61]. The latter is defined in terms of the generators $X_n, Z_n$, where $n = 1, \dots, L$, which satisfy $X_n^N = 1$, $Z_n^N = 1$, $Z_n X_n = \omega X_n Z_n$ where $\omega = \exp(2\pi i/N)$. In this case the Hamiltonian

$$H^{(N)} = H_0^{(N)} + h H_1^{(N)}, \tag{18}$$

$$H_0 = \sum_{n=1}^{L} \sum_{m=1}^{N-1} (1 - \omega^{-m})^{-1} X_n^m, \tag{19}$$

$$H_1 = \sum_{n=1}^{L} \sum_{m=1}^{N-1} (1 - \omega^{-m})^{-1} Z_n^m Z_{n+1}^{N-m}. \tag{20}$$

The generators of the Dolan-Grady relations (15) in this case are given by $A_0 = 4N^{-1}H_0$ and $A_1 = 4N^{-1}H_1$, where $H_{0,1}$ are defined in (18). This model has potential relevance to the parafermions in 1D [62].

### 3.3.2 Onsager algebra and the transfer matrix

To make contact with the next Section we point out a connection between Onsager algebra and the traditional transfer matrix approach for statistical system. Namely, while the transfer matrix of the Ising model

$$\mathcal{T}(\beta_1, \beta_2) = \exp(\beta_1 A_0) \exp(\beta_2 A_1) \tag{21}$$

does not commute with the integrals $Q_m$'s defined in (17), one can easily find a different set of integrals $I_m$ which do commute with $\mathcal{T}(\beta_1, \beta_2)$. These integrals are combinations of elements of the Onsager algebra and can be determined by considering commutativity condition $[\mathcal{T}(\beta_1, \beta_2), I_m] = 0$. If we rewrite this condition in the form of $e^{\beta_2 A_1} I_m e^{-\beta A_1} = e^{-\beta_1 A_0} I_m e^{\beta_1 A_0}$, then assuming certain expansion for $I_m = \sum_p a_p A_p + g_p G_p$ one can use the BCH formula to determine the coefficients $a_p$ and $g_p$ in terms of $\beta_{1,2}$. Since the operators $A_m$ and $G_m$ involve $m + 1$ consecutive spins on the lattice we write here only the lowest order in $m$ result,

$$I_0 = A \sum_n \sigma_n^x \sigma_{n+1}^x + B \sum_n \sigma_n^z \tag{22}$$

$$+ \; C \sum_n (\sigma_n^x \sigma_{n+1}^y + \sigma_n^y \sigma_{n-1}^x) \tag{23}$$

$$A = \cos(2JT_1)\sin(2hT_2), \tag{24}$$

$$B = \sin(2JT_1)\cos(2hT_2), \tag{25}$$

$$C = \sin(2JT_1)\sin(2hT_2). \tag{26}$$

This Hamiltonian, since it commutes with the $\log(\exp(\beta_1 A_0) \exp(\beta_2 A_1))$ has the same system of eigenstates as the Floquet Hamiltonian. Note that the complete Floquet Hamiltonian involves infinite number of commuting integrals of motion. The explicit form of these conserved charges was obtained in [63], [64]. However, since the Floquet Hamiltonian commutes with the Ising transfer matrix, it shares the same structure of the phase diagram and the same *critical indexes* at the phase transition line. We note that one can locate the phase transition

by self-duality. We mention that recently, periodically driven Ising-type chains have attracted a lot of attention as a viable platform for dynamical generating of critical and topological states [65], [66].

Comparing (21) with (1) it becomes clear that the Floquet Hamiltonian $H_F(T)$ is related to the log of the Ising model transfer matrix with analytically continued parameters $\beta_1$ and $\beta_2$,

$$H_F \sim \log \mathscr{T}(\beta_1, \beta_2), \tag{27}$$

$$\beta_{1,2} = -iT_{1,2}, \tag{28}$$

where $T_{1,2}$ are the driving periods (see Fig. (1) and Eq. (1)). The eigenvalues $\Lambda$ of the Ising transfer matrix are known for the finite-dimensional case [59]. From that the eigenvalues of $H_F$ are obtained by analytic continuation. These observations lead us next to consider the models related to the transfer matrices of integrable models.

# 4 Models related to solvable statistical mechanics

Here we focus on two more non-trivial classes of models. They are related to fundamental objects in statistical mechanics - transfer matrices. In the theory of 2D exactly solvable models of lattice statistical physics [67] Baxter defined two types of the transfer matrices: (i) the *row transfer matrix* (RTM), denoted as $\mathscr{T}_R(\lambda)$, and the (ii) *corner transfer matrix* (CTM), denoted as $\mathscr{T}_C(\lambda)$. The latter, in particular, is extremely useful in computing a one-point function of local spin variables of 2D lattice models. First we remind construction of integrable models and corresponding transfer matrices and then apply them to Floquet systems.

## 4.1 Row and corner transfer matrices

By the quantum-classical correspondence 1D integrable quantum chain models are equivalent to the 2D integrable lattice classical statistical systems, see e.g [68]. Let us briefly summarize some key notions of quantum integrable systems, which will become important later. By saying that the Hamiltonian $H_{\text{int}}$ is integrable we mean that this Hamiltonian can be derived from the transfer matrix $\mathscr{T}_R(\lambda)$, which is a product of local Lax operators $L_j(\lambda)$ defined at every lattice site $j$: $\mathscr{T}_R(\lambda) = \prod_{j=1}^N L_j(\lambda)$, where $N$ is the system size. Here $\lambda$ is a complex parameter called rapidity. Typically, the Lax operator can be defined as a matrix with operator-valued entries which satisfy a certain algebra. In the simplest case of the XXZ spin-1/2 chain the Lax operator is a $2 \times 2$ matrix with entries which belong to the spin-1/2 representation, e.g. $L_j^{(12)} \sim \sigma_j^-$, $L_j^{(21)} \sim \sigma_j^+$ while the diagonal elements are the functions of $\sigma^z$ [69]. The size of the Lax matrix defines dimension of the auxiliary space $d$. Therefore $\mathscr{T}_R(\lambda)$ can be viewed as a $d \times d$ matrix with operator entries, which in turn are complicated functions of the spin operators of the whole lattice. The Yang-Baxter integrability structure is fixed by the structure of the so-called $R$-matrix $R_{1,2}(\lambda, \mu)$ which intertwines two copies of $\mathscr{T}_R(\lambda)$ operators (indexes 1, 2 refer to two quantum spaces), $R_{1,2}(\lambda, \mu)\mathscr{T}_R(\lambda)\mathscr{T}_R(\mu) = \mathscr{T}_R(\mu)\mathscr{T}_R(\lambda)R_{1,2}(\lambda, \mu)$. Moreover, the $R$ matrix satisfies the famous Yang-Baxter equation

$$R_{1,2}(\lambda, \mu)R_{1,3}(\lambda, \nu)R_{2,3}(\mu, \nu) = R_{2,3}(\mu, \nu)R_{1,3}(\lambda, \nu)R_{1,2}(\lambda, \mu), \tag{29}$$

which is a consequence of the consistency of the $RTT = TTR$ relation above. Another consequence of the latter is that the traces of the transfer matrices over the auxiliary space

$$\tau(\lambda) = \text{Tr}_a \mathscr{T}_R(\lambda), \tag{30}$$

where $\text{Tr}_a$, $a = 1, \ldots d$, denotes trace over the auxiliary space, commute for different values of the spectral parameters $\lambda, \mu$, $[\tau(\lambda), \tau(\mu)] = 0$. This commutativity and therefore existence of the $R$-matrix ensures that there is a set of $N$ commuting conserved operators $Q_n$, $n = 0, \ldots N$ such that the local physical Hamiltonian $H_{\text{int}}$ is usually chosen to be $Q_1$, while $Q_0$ is identified with total momentum. Here the label $n$ corresponds to the $n$-th derivative of the logarithm of the transfer matrix with respect to $\lambda$. These integrals are local (i.e. have a local support on a lattice) and are mutually commuting because matrices $\tau$ commute at different $\lambda$, namely

$$\log \tau(\lambda) \sim \sum_{n=1} \frac{\lambda^n}{n!} Q_n. \tag{31}$$

Another property which follows from integrability is the existence of the so-called *boost* operator $B$ which we are going to discuss now.

In a 1D quantum formulation of the 2D statistical model, the RTM is nothing but the transfer matrix $\mathscr{T}_R(\lambda)$ introduced above, which satisfies the Yang-Baxter equation, while the CTM $\mathscr{T}_C(\lambda)$ is acting on $\mathscr{T}_R(\lambda)$ by a shift of the spectral parameter as follows

$$\tau(\lambda + \mu) = \mathscr{T}_C^{-1}(\mu)\tau(\lambda)\mathscr{T}_C(\mu). \tag{32}$$

The CTM can be shown to satisfy a group property, $\mathscr{T}_C(\mu)\mathscr{T}_C(\lambda) = \mathscr{T}_C(\mu + \lambda)$. This allows to write CTM as

$$\mathscr{T}_C(\lambda) = \exp(-\lambda B), \tag{33}$$

where $B$ is called the boost operator. In a sense it plays the role of the CTM Hamiltonian,

$$B \equiv H_{CTM}. \tag{34}$$

In terms of $B$, Eq. (32) takes a differential form

$$[B, \tau(\lambda)] = \frac{\partial}{\partial \lambda} \tau(\lambda). \tag{35}$$

For the lattice models, substituting expansion (31), we observe that application of the boost operator generate new conserved quantities from the old ones [70]

$$[B, Q_n] = iQ_{n+1}. \tag{36}$$

(we assume that all $Q_n$'s are Hermitian). The CTM plays crucial role in relating six vertex model (quantum XXZ model) with quantum affine algebras [71–74]. This connection implies that the spectrum of the CTM defined on half of the 1D chain belongs to a representation of the quantum algebra $U_q(\hat{sl}_2)$. Moreover, it was suggested in [75] that the eigenstates of $B$ on the whole chain are given by the Fourier transform of the Bethe eigenstates. The eigenvalues of the boost operator are parametrized by integer numbers $j = 0, 1, \ldots$ with the zero eigenvalue corresponding to the ground state and a single parameter $\epsilon$, which is a function of the model's parameters. This, in particular implies that $\mathscr{T}_C(2\pi i/\epsilon)$ is an identity so $\tau(\lambda + 2\pi i/\epsilon) = \tau(\lambda)$. While the proof of this statement is not known to us for the general case of XXZ, it is supported by numerical computations in the Ising and XXZ cases [76,77] and by the Baxter's conjecture [67] that there is an intertwining operator $I$, which transforms the spectrum of CTM into the one for the Ising model. Summarizing these, the spectrum of the boost operators is of the Ising type [78]

$$H_{CTM} = \sum_{j=0}^{\infty} \epsilon_j n_j, \tag{37}$$

where $n_j = 0, 1, 2\ldots$ is an integer and

$$\epsilon_j = \begin{cases} (2j+1)\epsilon & \text{for} \quad \gamma < 1 \\ 2j\epsilon & \text{for} \quad \gamma > 1. \end{cases} \tag{38}$$

Here $\gamma$ is defined in Eq. (16). For the transverse field Ising model

$$\epsilon = \pi \frac{K(\sqrt{1-k^2})}{K(k)}, \quad k = \min[\gamma, \gamma^{-1}], \tag{39}$$

where $K(k)$ is the complete elliptic integral of the first kind. For the anisotropic (XXZ) Heisenberg model (considered below) the spectrum has the same form with

$$\epsilon_j = 2j\epsilon, \qquad \epsilon = \text{arccosh}\Delta, \tag{40}$$

where the anisotropy parameter is $\Delta > 1$. It is interesting to note that the lattice version of the Virasoro algebra is constructed using the boost operator with $B \sim L_0$ [79, 80] . While as we mentioned we are not aware of a general proof that the spectrum of the Boost operator always has a linear dispersion (37), there are no counterexamples showing otherwise. Therefore we will use this assumption to make some general statements about Floquet boost models below. Note also, that some spin chains at criticality may have a quadratic spectrum as well [81].

If we represent our integrable lattice Hamiltonian $H_{\text{int}} \equiv Q_2$ in terms of the local densities $h_{ij}$ as $H_{\text{int}} = \sum_j h_{j,j+1}$ the generic form of the boost operator is then [70]

$$B = \sum_j j h_{j,j+1}. \tag{41}$$

In general (for example in the presence of magnetic field) the above expression should be modified by a local term: $\sum_j h'_j$, with some local $h'$. Since $[Q_n, Q_m] = 0$, $\forall m, n$ one can in principle consider a "Hamiltonian"

$$H_{eff} = \sum_{n=2}^{\infty} a_n Q_n + bB \tag{42}$$

for some parameters $a_n$ and $b$. We will show below that this is a generic form of the Floquet Hamiltonian for this class of models.

In the continuum (field theory) limit the boost $B$, the Hamiltonian $H$ and the momentum $P$ operators form a Poincare algebra [82], [83],

$$[H, P] = 0, \quad [B, P] = iH, \quad [B, H] = iP. \tag{43}$$

It is interesting that the continuum limit from the lattice conserved charges goes as follows: $Q_{2k} \sim H$ while $Q_{2k+1} \sim P$ for any integer $k$.

The boost operator plays an important role in computing the *entanglement entropy* in recent studies of this object in field theory [84, 85]. In fact the Hamiltonian $H_{CTM} = B$ is identified with the *entanglement Hamiltonian* [86]. The same interpretation applies to the lattice models where it has been used for DMRG-based studies of entanglement [78, 87–89]. Note that the relation between the entanglement Hamiltonian and corner Hamiltonian in critical spin chains has been intensively studied recently [90] for a broad class of $SU(N)$ symmetric spin chains.

## 4.2 Floquet models related to the row transfer matrix

The driven Ising model from the previous section is a particular example of much more general class of models. In fact, transfer matrices $T$ of majority (if not all) classical two-dimensional solvable statistical mechanical models can be represented in the following form (see chapters 6 and 7.2 in Ref. [67] and also Ref. [91]):

$$\mathcal{T}_R = \underbrace{VWVW\ldots VW}_{M} \,, \tag{44}$$

where $V$ is the transfer matrix which adds a row of horizontal edges to the square lattice and $W$ adds a row of vertical edges to the same lattice. There are $M$ products of $VW$ in (44). Each of the matrices $V$ and $W$ have the following structure,

$$
\begin{aligned}
V &= X_1 X_3 X_5 \ldots X_{2N-1}, \\
W &= X_2 X_4 X_6 \ldots X_{2N},
\end{aligned} \tag{45}
$$

where $N$ is even. This construction applies, in particular to the square lattice model of $M$ rows and $N/2$ columns, where $X_{2j-1}$ is a local transfer matrix that adds to the lattice a vertical edge in column $j$ while $X_{2j}$ is a matrix which adds a horizontal edge between columns $j$ and $j+1$. For the lattice models for which local variables take $q$ values (e.g. $q = 2$ for the Ising model), these matrices are of dimension $q^{N/2}$. It is assumed here that matrices $X_j \equiv X_j(x)$ are the functions of some parameters $x$ related to the Boltzmann weights of the model.

The key property of matrices $X_j(x)$ is that they satisfy the Yang-Baxter equation, see Ref. [91] and also Chapter 12.4 in Ref. [67],

$$X_j(x)X_{j+1}(x')X_j(x'') = X_{j+1}(x'')X_j(x')X_{j+1}(x) \tag{46}$$

where $X_j(x)$ can be identified with $\mathscr{P}R_{12}(\lambda-\mu)$ from the algebraic Bethe ansatz introduction to this section, where $\mathscr{P}$ is the permutation operator acting between two (quantum) spaces. This ensures integrability of the lattice statistical model.

For a broad class of models the operators $X_j$'s can be chosen in the following form, see Ref. [91] (up to an overall multiplication factor),

$$X_{2j-1} = 1 + x_1 U_{2j-1}, \qquad X_{2j} = 1 + x_2 U_{2j}, \tag{47}$$

where $x_{1,2}$ are two independent parameters. Here the matrices $U_j$ satisfy so-called Temperley-Lieb algebra,

$$
\begin{aligned}
U_j^2 &= Q U_j, \\
U_j U_{j+1} U_j &= U_j, \\
U_j U_k &= U_k U_j, \qquad |j-k| \geq 2
\end{aligned} \tag{48}
$$

where $Q$ is a number (we assume it to be real here). Because of this algebra the matrices $V$ and $W$ can be put into exponential forms,

$$
\begin{aligned}
V &= \exp\left[\alpha(U_1 + U_3 + U_5 + \ldots + U_{2n-1})\right], \\
W &= \exp\left[\beta(U_2 + U_4 + U_6 + \ldots + U_{2n})\right],
\end{aligned} \tag{49}
$$

such that

$$x_1 = Q^{-1}[\exp(\alpha Q) - 1], \quad x_2 = Q^{-1}[\exp(\beta Q) - 1]. \tag{50}$$

Representation theory of the Temperley-Lieb algebra is well developed. We mention here a few interesting cases. For the lowest-dimensional representations in terms of the Pauli matrices two cases are known. Defining $Q = q + q^{-1}$ we have for $q = \exp(i\pi/4)$ the Ising-like representation,

$$U_{2j}^{(I)} = \frac{1}{\sqrt{2}}\left(1 + \sigma_j^z \sigma_{j+1}^z\right),$$ (51)

$$U_{2j-1}^{(I)} = \frac{1}{\sqrt{2}}\left(1 + \sigma_j^x\right).$$ (52)

The second one is related to the $XXZ$ model and is given by

$$U_j^{(XXZ)} = -\frac{1}{2}\left[\sigma_j^x \sigma_{j+1}^x + \sigma_j^y \sigma_{j+1}^y + \cos(\eta)(\sigma_j^z \sigma_{j+1}^z - 1) + i\sin(\eta)(\sigma_j^z - \sigma_{j+1}^z)\right],$$ (53)

where $q = \exp(i\eta)$. As we are focusing on Hamiltonian Floquet systems the operators $U_j$ should be Hermitian ensuring that the evolution operators $V$ and $W$ are unitary. Thus we have to assume that the parameter $\eta$ is purely imaginary, $\eta = i\kappa$. If we set $\eta = 0$ (note that another possible choice of $\eta = \pi$ results in an equivalent model) we obtain the isotropic Heisenberg form for the Temperley-Lieb generators $U$'s

$$U_j^{(XXX)} = -\frac{1}{2}\left[\sigma_j^x \sigma_{j+1}^x + \sigma_j^y \sigma_{j+1}^y + (\sigma_j^z \sigma_{j+1}^z - 1)\right].$$ (54)

For imaginary $\eta$ one can, in principle, generate anisotropic spin chain with a special form of magnetic field (namely, the Hamiltonians $H_{1,2}$ would have a staggered magnetic fields). Here we focus on the isotropic case, however.

We thus see that in the Ising case the Hamiltonian $H_1$ of the Floquet protocol can be identified with $\sum_j U_{2j-1}^{(I)}$, $H_1 \equiv \sum_{j=1} U_{2j-1}^{(I)}$ which is the $z - z$ coupling term of the transverse Ising model while the $H_2 = \sum_{j=1} U_{2j}^{(I)}$ is proportional to the traverse field part of the Ising Hamiltonian. Moreover the next term of the BCH expansion generates the term Eq.(23) in the expansion for $I_0$, and so on. We are thus obtaining the same type of the protocol and the corresponding Floquet Hamiltonian as in the previous Section. This similarity is a peculiarity of the Ising model (probably rooted in its super-integrability). On the other hand the corresponding Floquet system related to the XXX model is a new one. The Floquet protocol consists of switching between even and odd links of the uniform Heisenberg model and the Hamiltonians $H_{1,2}$ of the Floquet protocol are defined as

$$H_1^{XXX} \equiv \sum_{j=1}^N U_{2j-1}^{(XXX)},$$

$$H_2^{XXX} \equiv \sum_{j=1}^N U_{2j}^{(XXX)}.$$ (55)

It is interesting that the Floquet integrability ensures that we are getting an integrable model in each order of $BCH$ expansion (2). For example setting $\alpha = \beta = iT/2$ in Eq. (49) we find that in the leading order in this (high frequency) expansion

$$H_F^0 = \frac{1}{4}\sum_j \vec{\sigma}_j \cdot \vec{\sigma}_{j+1}$$ (56)

is just the standard Heisenberg model. In the next leading order

$$H_F^{0+1} = H_F^0 + \frac{T}{4}\sum_j (\vec{\sigma}_j \times \vec{\sigma}_{j+1}) \cdot \vec{\sigma}_{j+2},$$ (57)

Here, the second term of the expansion is proportional to the conserved charge $Q_3$ of the isotropic Heisenberg model.

Thus summarizing, in order to construct integrable Floquet Hamiltonians related to the RTM we need to identify $V$ with the evolution exponent of $H_1 = U_1 + U_3 + \dots U_{2n-1}$ and the operator $W$ with the evolution exponent of $H_2 = U_2 + U_4 + \dots U_{2n}$. The constants $\alpha = -iT_1$ and $\beta = -iT_2$ are identified with the time intervals for $H_1$ and $H_2$. In the language of statistical mechanics this corresponds to the lattice model with the *complex Boltzmann weights* $x_{1,2}$ defined in (50). If the analytic continuation to the complex domain can be done safely, we are getting an integrable Floquet system.

Physical interpretation of the Floquet protocol goes as follows. In the case of the Heisenberg model the system which would realize the integrable protocol is defined as a periodic modulation (switching "on" and "off") of even and odd links of the spin chain. The same protocol of switching between even and odd links has been suggested before in [92] as a way to generate long-range spin entanglement and resonating valence bond spin liquid in a double well ultracold atomic superlattices.

In the Ising case this protocol corresponds to a periodic switching between the transverse field and the Ising interaction. It is interesting that for Floquet integrable models there is no need to send the driving period to zero with increasing system size as e.g. is required in generic implementation of digital quantum simulation to avoid heating (see e.g. Ref. [40]). Thus implementing integrable Floquet protocols can serve as a guide of developing stable Trotterization schemes and hence stable digital quantum simulators.

We note that the line $x_1 x_2^{-1} = 1$ is self-dual (in a sense of the Kramers-Wannier duality, see [67]) and corresponds to the critical CFT models [93]. Whether this survives after the Wick rotation is, to our knowledge, an open problem. One more interesting aspect is that this class of models has quantum $U_q(sl_2)$ symmetry. At the special point

$$T_1 Q = \pi + 2\pi n, \quad T_2 Q = \pi + 2\pi m, \tag{58}$$

where $n, m$ are integers, the Boltzmann weights $x_{1,2}$ become real again. In Ref. [93] the phase diagram of these vertex models with real $\exp(\alpha Q)$ has been studied for $x_1 x_2^{-1} = 1$ (in our notations). Applying these results one can observe that when the self-duality condition is satisfied, the condition (58) leads to the horizontal line on a phase diagram of [93–95]. This line meets a critical "antiferromagnetic line" at $Q = 2$ ($Q = 4$ in the notations of [93]). This formally corresponds to the Conformal Field Theory with the central charge $c = -\infty$. We note that $x_1 x_2^{-1} = 1$ for (58) when $Q = 2$. On the other hand $Q = 2$ is the case of the isotropic Heisenberg chain protocol, suggesting that the protocol (55) realizes a critical Floquet system.

As we can see, in this class of models the operators $X_j(x)$, after an appropriate Wick rotation, are identified with the local evolution operator of the quantum problem. Their alternating product represent a Floquet evolution operator. One can ask a more general question, namely what are the solutions of the condition on operator $X_j(x)$ to satisfy the Yang-Baxter relation and to have an exponential form (to ensure the local semigroup relation $X_j(x)X_j(y) = X_j(x+y)$)? This question has been addressed in Refs. [96, 97], where several classes of solutions have been identified. It would be interesting to construct corresponding Floquet systems associated with these solutions.

## 4.3 Floquet models related to the corner transfer matrix: "boost models"

Here we focus on a class of models generated by the boost operator generating CTMs. Specifically we consider a model defined in such a way that one of the parts of the Floquet system, namely $H_2$ is an integrable (in the Yang-Baxter sense) lattice Hamiltonian, $H_2 \equiv H_{\text{int}}$, while $H_1$ is proportional to the corresponding boost operator, $H_1 = bB$. Lets analyze the BCH formula

(2) and identify $X \equiv -iT_1 B$ and $Y \equiv -iH_{\text{int}}T_2$. By looking at the structure of (2) it becomes clear that, according to (36) in the $n$-th order of the expansion we will be generating new integrals of motion $Q_n$ coming from the commutator of the $Q_{n-1}$ with the Boost operator. It is clear that the only part of the BCH formula which survives contains the original Hamiltonians $H_1 \sim B$, $H_2 = H_{\text{int}}$ and the nested commutators

$$[B, [B, \dots [B, H_2] \dots ]] \tag{59}$$

of arbitrary length. Then, clearly, the Floquet Hamiltonian will be given by the form (42) above with

$$a_n = \frac{B_{n-2}}{(n-2)!} \frac{T_1^{n-2} T_2}{T}, \qquad n \geq 2, \tag{60}$$

where $B_n$ are Bernoulli numbers. Some first nonzero $B_n$'s from the so-called second sequence (exactly those which enter the BCH formula) are $B_0 = 1, B_1 = \frac{1}{2}, B_2 = \frac{1}{6}, B_4 = -\frac{1}{30}, B_6 = \frac{1}{42}, \dots$. By construction, the effective *exact* Floquet Hamiltonian (42) is integrable. Using the generating function for Bernoulli's numbers one can, in principle, convert an expression for $H_F$ into the form of the integral transform. However, the convergence of this formal expression should be checked for every state $|\Psi_0\rangle$ separately. For this reasons we avoid presentation of these formal expressions. In Appendix B we demonstrate convergence of the expectation value of the Hamiltonian (42) for several initial product states $|\Psi_0\rangle$ and two particular protocols $\lambda(t)$.

Another problem with this effective representation is that while the first term is diagonal in the basis of Bethe states, it is not clear at this point how to deal with the boost term. Note also that direct evaluation of the expectation value for the boost operator, $\langle \Psi_0 | B | \Psi_0 \rangle$ leads to the divergent result in the thermodynamic limit. This divergence stems from the fact that the boost operator is similar to a uniform electric field. However, in integrable dynamical systems we are considering here periodic application of the Boost operator does not lead to heating and thus to divergencies in physical observables. For the Boost models the results of this section can be extended to more general driving protocols by going to the rotating frame, which we will discuss next.

*Rotating frame*

One can start first with somewhat more general time-dependent Hamiltonian,

$$H = H_{\text{int}} + b(t)B \tag{61}$$

and assume that $b(t)$ is a periodic function of time $b(t + T) = b(t)$. Then by construction we are dealing with a Floquet problem. We note that we can also consider an arbitrary function $b(t)$ in the interval $[0, T]$, where $T$ is the time of interest at which we want to evaluate observables, and then periodically continue it for $t > T$. So our general results will equally apply to FLoquet systems, quantum quenches or any other arbitrary time dependences. It is convenient to rewrite the Hamiltonian as

$$H = H_{\text{int}} + \bar{b}B + \delta b(t)B, \tag{62}$$

where

$$\bar{b} = \frac{1}{T} \int_0^T b(t)dt \qquad \mod \quad \frac{2\pi}{T\epsilon},$$

$$\delta b(t) = b(t) - \bar{b}, \tag{63}$$

where $\epsilon$ is the parameter defined below Eq. (37) Note that $\delta b(t)$ is also a periodic function of time. Here we use the fact that the spectrum of the boost operator is proportional to integer numbers, see Eqs. (38), (40).

Next let us go to the rotating frame with respect to the last, time-dependent term generated by the unitary

$$V(t) = \exp(-iF(t)B),$$
$$F(t) = \int_{t_0}^{t} \delta b(t')dt'.$$
(64)

We note that by construction $F(0) = 0$ and for integer $m$, $F(mT) = 2\pi n m/\epsilon$, where $n$ is also an integer defined by

$$\int_0^T b(t)dt = T\bar{b} + \frac{2\pi n}{\epsilon}.$$
(65)

Then, using that all eigenvalues of $B$ are integer multiples of $\epsilon$, we see that

$$V(mT) = \exp[-2\pi i n m B/\epsilon] = \hat{I}$$
(66)

is the unity operator such that $V(t)$ is a periodic function of time. Thus the transformation to the rotating frame does not break periodicity in time. The Hamiltonian in the rotating frame is [98]

$$H_{rot} = V^\dagger(t)HV(t) - iV^\dagger(t)\partial_t V^\dagger = V^\dagger(t)(H_2 + \bar{b}B)V(t) = \bar{b}B + V^\dagger(t)(H_2)V(t)$$
(67)

for generic $H$. When the Taylor series for the last expression converges in the operator sense the rotating frame Hamiltonian in our case is equivalent to

$$H_{rot} = \bar{b}B + \sum_{n=1}^{\infty} \frac{[F(t)]^{n-1}}{(n-1)!}Q_n.$$
(68)

Here we identify $Q_1 \equiv H_{\text{int}}$. The last expression is formally related (see Eq. (31)) to the derivative of the RTM $\tau(\lambda)$

$$H_{rot} = \bar{b}B + \partial_\lambda \log \tau(\lambda)|_{\lambda=F(t)},$$
(69)

where $F(t)$ is defined in (64). The question of convergence of this formal sum for arbitrary time should be considered separately. We discussed several convergent examples in Appendix B.

A particularly simple expression for the Floquet Hamiltonian and hence the evolution operator over the period appears when $\bar{b} = 0$ i.e. when

$$\int_0^T b(t)dt = \frac{2\pi n}{\epsilon},$$
(70)

where $n$ is an (arbitrary) integer. In this case, the Floquet Hamiltonian is simply equal to the time average of $H_{rot}$:

$$H_F = \overline{\partial_\lambda \log \tau(\lambda)|_{\lambda=F(t)}},$$
(71)

where the over line denotes the period averaging. This follows e.g. by observing that all terms in $H_{rot}$ commute with each other at different times and thus one can omit time-ordering in the evolution operator

$$U(T) = \exp\left[-i\int_0^T H_{rot}(t)dt\right] = \exp[-iT\bar{H}_{rot}].$$
(72)

The Floquet Hamiltonian in this case is thus explicitly written as the sum of integrals of motion.

In particular, there are two interesting classes of driving protocols for which the condition $\bar{b} = 0$ above is satisfied:

- Periodic Floquet protocols: $b(t) = b_0 \cos(\omega t)$. Then $\bar{b} = 0$ when

$$T_n = 2\pi n/\omega \tag{73}$$

  with an arbitrary integer $n$.

- Quench protocols $b(t) = b_0 \theta(t)$, where $\theta(t)$ is the step function. Then $\bar{b} = 0$ at

$$T_n = \frac{2\pi n}{\epsilon b_0}. \tag{74}$$

At these special times (73), (74) the energy of the system as well as all other conserved quantities exhibit full many-body revivals irrespective of the initial state of the system $|\Psi_0\rangle$

$$\langle \Psi(T_n)|Q_n|\Psi(T_n)\rangle = \langle \Psi(0)|Q_n|\Psi(0)\rangle. \tag{75}$$

In particular at these times the driving protocol performs exactly zero work on the system. It is interesting that the wave function does not necessarily return to itself. Indeed if we expand the wave function in the basis of the eigenstates of all integrals of motion, each component will generally acquire a different phase. For this reason if the system is initialized in a pure state, which is not an eigenstate of the Hamiltonian, the observables, which do not commute with the Hamiltonian will not be periodic in time. On the other hand if the system is initialized in an equilibrium state of the Hamiltonian, e.g. in an eigenstate of $H$ or in the Generalized Gibbs Ensemble then after times $T_n$ there will be complete revivals of the initial density matrix. Indeed in any equilibrium state phases of the wave functions are random and it does not matter if they are periodic or not. It is highly plausible that after long time any wave function at these discrete times will relax to the Generalized Gibbs Ensemble as in standard quench protocols but this question requires more careful investigation. Let us note that for the quench protocol these revivals at times given by Eq. (74) generalize Bloch oscillations to more generic integrable systems. The revivals are reminiscent of "quantum time crystals" actively discussed in the literature (see e.g. Ref. [99].) However, we can not escape from the similarity of the time-periodic state after a quench with an ordinary clock or any other frequency generator like a laser or maser, which have much longer history than time crystals. To avoid any injustice we decided to term this periodic state as a "Quantum Boost Clock" (QBC). It is clear that QBC can be extended to other driving protocols creating various periodic and aperiodic revivals at times satisfying Eq. (70). While strictly speaking Boost operators only exist in integrable models, it is intuitively clear that weak integrability breaking can only induce small heating such that the oscillating phases can exist for long but finite times. Moreover it is plausible that the heating can be reduced further by using the ideas of the counter-diabatic driving to find approximate local Boost operators [100]. Then QBC regimes will be analogous to prethermalized time crystals or simply to cuckoo clocks, which can only run for a finite amount of time determined by the gravitational energy stored in the weights and the energy dissipation in the clock.

## 5 Discussions and conclusions

In this paper we discussed and identified three classes of integrable Floquet models. The first class is defined by the Hamiltonians which are the linear combinations of the generators of

some classical Lie algebras. Possible Floquet systems in this case are in one-to-one correspondence with different conjugacy classes of Cartan subalgebras (or series of representations). In this way one can provide an algebraic classification of Lie-algebraic Floquet systems. Extensions of this class of models are related to infinite-dimensional algebras and algebras of a more complicated structure. The second and the third classes of models are related to classical lattice statistical mechanical models defined either by the row-to-row transfer (RTM) matrix or by the corner transfer matrix (CTM). While RTM-models are defined via Wick's analytic continuation of their Boltzmann weights, the CTM-models are defined through the boost operator, which in known cases coincides with the entanglement Hamiltonian.

We expect that our models contain interesting physics which will be studied elsewhere. It could be as much complicated as the physics of traditional equilibrium integrable models. In particular, it would be interesting to apply the recently obtained results for the XXZ spin chain for which the generating function of commuting integrals of motion has been computed explicitly [101, 102] for several relevant initial states. Quench dynamics similar to the one studied in [103], [104], [105], [106] is also possible to implement in the Floquet context, and we can also expect deviations from the predictions given by the Generalized Gibbs Ensemble. We hope in the future to address the physics of Floquet evolution from these initial states. We also showed that the class of Boost models can be extended to generic non-periodic protocols, in particular, quenches, where one can realize interesting phenomena like exact energy revivals, which realize QBC. Interestingly, boost models bear many parallels with recent work [38], where it was suggested that in related setups the system's wave function after a quench can follow the ground state of some local time-dependent Hamiltonian. We expect that other classes of integrable Floquet systems are possible. It would be interesting to find those which have no equilibrium counterparts.

## Acknowledgements

Both authors are grateful to the Newton Institute for Mathematical Sciences in Cambridge (UK) and to the organizers of the program "Mathematical Aspects of Quantum Integrable Models in and out of Equilibrium" (11.01.2016 - 05.02.2016) where the idea of this work has emerged. The work of V.G. is part of the Delta-ITP consortium, a program of the Netherlands Organization for Scientific Research (NWO) that is funded by the Dutch Ministry of Education, Culture and Science (OCW). A.P. was supported by NSF DMR-1506340, ARO W911NF1410540 and AFOSR FA9550-16-1-0334. We would like to thank Mikhail Pletyukhov for useful discussions and Andrei Zvyagin, Hosho Katsura and Tony Apollaro for useful correspondence.

## A    Conjugacy classes of Cartan subalgebras of real Lie algebras

Kostant and Sugiura developed classification of conjugacy classes of Cartan subalgebras of real Lie algebras. While in the case of complex Lie algebras there is unique Cartan subalgebra, the case of real Lie algebras is much more involved. The following table summarize the number of Cartans for different classical Lie algebras:

Consider, for example, the symplectic case, $\Lambda \in Sp(2n, \mathbb{R})$. In this case one can introduce a symplectic structure $\Omega_{kl}$. Different type of orbits of maximally abelian subalgebras correspond either to (i) hyperbolic (direct and inverse), (ii) elliptic, (iii) loxodromic and (iv) parabolic types. According to the Table 1 there are $(n+2)^2/4$ different Cartan subalgebras in the case of even $n$ and $(n+1)(n+3)/4$ in the case of $n$ odd. Every Cartan subgroup $\Lambda_{k,s} = \exp(h_{k,s})$ is parametrized by two numbers $(k,s)$ such that $k \geq 0, s \geq 0$ and $k + 2s \leq n$. The matrix $h_{k,s}$ has

| class | number of Cartans |
|---|---|
| AI, $A_l(B_{\frac{l}{2}})$ | $[\frac{l}{2}]+1$ |
| AII, $A_l(C_{\frac{l+1}{2}})$ | 1 |
| AIII, $A_l(A_{j-1} \oplus A_{l-j} \oplus D_1)$ | $j+1$ |
| BI $B_l(B_{\frac{2l-j}{2}} \oplus D_{\frac{j}{2}})$, $j$ even | $\frac{(j+2)^2}{4}$ |
| BI $B_l(B_{\frac{j-1}{2}} \oplus D_{\frac{2l-j}{2}})$, $j$ odd | $\frac{(j+1)(j+3)}{4}$ |
| DI, $D_l(D_{\frac{j}{2}} \oplus D_{\frac{2l-j}{2}})$, $j$ even | $\frac{1}{2}([l/2]+1)([l/2]+2)$ |
| DI, $D_l(B_{\frac{j-1}{2}} \oplus B_{\frac{2l-j-1}{2}})$, $j$ odd | $\frac{1}{2}([m/2]+1)([m/2]+2)$ |
| DIII, $D_l(D_1 \oplus A_{l-1})$ | $[\frac{l+2}{2}]$ |
| CI, $C_l(D_1 \oplus A_{l-1})$, $l$ even | $\frac{(l+2)^2}{4}$ |
| CI, $C_l(D_1 \oplus A_{l-1})$, $l$ odd | $\frac{(l+1)(l+3)}{4}$ |
| CII, $C_l(C_j \oplus C_{l-j}$ | $j+1$ |

Table 1: Number of conjugacy classes of Cartan subgroups according to Cartan classification.

a general form [52]

$$\begin{pmatrix} 0 & 0 & 0 & A_k & 0 & 0 \\ 0 & B_s & 0 & 0 & 0 & 0 \\ 0 & 0 & C & 0 & 0 & 0 \\ -A_k & 0 & 0 & 0 & 0 & 0 \\ 0 & 0 & 0 & 0 & -B_s & 0 \\ 0 & 0 & 0 & 0 & 0 & -C \end{pmatrix}, \tag{76}$$

where diagonal block matrices $A, B, C$ are defined as $A_k = \mathrm{diag}(h_1, h_2, \ldots, h_k)$, $C = \mathrm{diag}(h_{k+2s+1}, h_{k+2s+2}, \ldots, h_n)$, $B_l = \mathrm{diag}(b^{(1)}, b^{(2)}, \ldots, b^{(l)})$, where

$$b^{(r)} = \begin{pmatrix} h_{k+s+r} & -h_{k+r} \\ h_{k+r} & h_{k+l+r} \end{pmatrix}. \tag{77}$$

Here $h_j$, $(j = 1, \ldots n)$ are the real numbers. The matrix $\Lambda$ has therefore $2k$ eigenvalues of the unit circle type $\lambda_m = \exp(\pm i h_m)$ $(m = 1, \ldots, k)$ which define the elliptic blocks, then $2(n-k-2s)$ real eigenvalues $\lambda_j = e^{\pm h_j}$, $(k+2s+1 \leq j \leq n)$ defining the hyperbolic blocks, and $4l$ complex eigenvalues $\lambda_r = \exp(\pm i h_{k+r} \pm h_{k+s+r})$, $r = 1, \ldots l$ which define loxodromic block. Note that if only elliptic blocks are present this correspond to the maximal compact Cartan subgroup, while the opposite case of only hyperbolic blocks correspond to the maximally non-compact case. Parabolic blocks are obtained by considering nilpotent subalgebras. For example $(2 \times 2)$ parabolic blocks are generated by the nilpotent matrix

$$P = \pm \begin{pmatrix} 0 & 1 \\ 0 & 0 \end{pmatrix} \to e^{\kappa t P} = \begin{pmatrix} 1 & \kappa t \\ 0 & 1 \end{pmatrix}, \tag{78}$$

where $\kappa$ is a constant.

There is a connection between unitary representations and abelian subgroups. As we have seen in the main text, the elliptic blocks correspond to the discrete series of representations while the hyperbolic and parabolic blocks correspond to continuous series.

It would be interesting to construct Floquet systems corresponding to cases of the Lie algebras different than $sp(2n, R)$.

We also note that this algebraic analysis in terms of subalgebras and block is very important for stability analysis of system under nonlinear perturbation. This, in particular, is a subject of the periodic orbit theory. In the semiclassical approach based on the Gutzwiller trace formula, hyperbolic and loxodromic blocks characterize unstable directions of periodic orbits. Parabolic blocks are marginally unstable and exhibit a linear growth of the perturbation along the direction spanned by the eigenvector. Elliptic blocks describe stable motion under perturbation if all the eigenvalues are mutually irrational (for review see [56]). These and many other results constitute the Krein-Gelfand-Lidskii-Moser theory of structural stability under perturbations. Under a perturbation the parabolic block bifurcates into a hyperbolic ($\kappa > 0$) or elliptic ($\kappa < 0$) blocks. The Morse index is a topological invariant which can be used to quantify stability of the Floquet dynamics under nonlinear perturbation.

## B  Matrix elements of Floquet Hamiltonian in the product states for the boost class of models: example of the $XXZ$ spin chain

The XXZ model is a model for anisotropic spin chain

$$H_{XXZ} = \frac{J}{4} \sum_{j=1}^{L} (\sigma_j^x \sigma_{j+1}^x + \sigma_j^y \sigma_{j+1}^y + \Delta(\sigma_j^z \sigma_{j+1}^z - 1)). \tag{79}$$

Eigenstates and eigenvalues are described by the system of Bethe ansatz equations. They can be solved only numerically. Many things drastically simplify in the thermodynamic limit. Therefore from now on we proceed with $L \to \infty$ and with a gapped regime $\Delta > 1$.

The generating function of commuting integrals of motion has been computed explicitly [101], [102] for several relevant initial states and can be directly used in our context. Namely, using the notations and conventions for the generating function of [101, 102], which is obtained from the logarithmic derivative of the trace of the transfer matrix

$$\Omega_{\Psi_0}(\lambda) = -i \sum_{k=1}^{\infty} \left(\frac{\eta}{\sinh \eta}\right)^k \frac{\lambda^{k-1}}{(k-1)!} \frac{\langle \Psi_0 | Q_k | \Psi_0 \rangle}{L}, \tag{80}$$

where $\Delta = \cosh \eta$. Here we identify $Q_1$ as a physical Hamiltonian. It is easy to see that it is obviously related to the expectation value of our Hamiltonian in the rotating frame (68). In particular, for a given initial state $|\Psi_0\rangle$

$$\langle \Psi_0 | H_{rot} | \Psi_0 \rangle = \left(\frac{\sinh \eta}{\eta}\right) \Omega_{\Psi_0}(\lambda)|_{\lambda=\lambda^*} + \bar{b} \langle \Psi_0 | B | \Psi_0 \rangle, \tag{81}$$

$$\lambda^* = F(t) \frac{\eta}{\sinh \eta}. \tag{82}$$

This implies that, assuming the convergence of the series expansion in (68) (which allows for exchange of summation and time average), the Floquet Hamiltonian is

$$\langle \Psi_0 | H_F | \Psi_0 \rangle = i \frac{\sinh \eta}{\eta} \frac{1}{T} \int_0^T \Omega_{\Psi_0}\left(F(t) \frac{\sinh(\eta)}{\eta}\right) dt \tag{83}$$

provided that the condition $\bar{b} = 0$ has been met. Explicit expressions for several initial states $|\Psi_0\rangle$ were computed in [101] and in [102]. We focus here on three interesting examples:

1. Ferromagnet in $x$-direction,
   $|x, \uparrow\rangle = \otimes_{j=1}^{L} \frac{1}{\sqrt{2}}(|\uparrow\rangle_j + |\downarrow\rangle_j)$

$$\Omega_{x,\uparrow}(\lambda) = \frac{i\eta \sinh(\eta)}{2 + 2\cos(\eta\lambda) + 4\cosh(\eta)}; \tag{84}$$

2. Neel state in $z$ direction,
   $|N\rangle = \frac{1}{\sqrt{2}}(|\uparrow\downarrow\uparrow \ldots + |\downarrow\uparrow\downarrow \ldots\rangle$

$$\Omega_N(\lambda) = \frac{i\eta \sinh(2\eta)}{2\cosh(2\eta) + 2 - 4\cos(\eta\lambda)}; \tag{85}$$

3. Dimer (Majumdar-Ghosh) state
   $|D\rangle = \prod_{j=1}^{L/2} \frac{1}{2}(|\uparrow\rangle_{2j-1}|\downarrow\rangle_{2j} - |\downarrow\rangle_{2j-1}|\uparrow\rangle_{2j})$

$$\Omega_D = \frac{i\sinh(\eta)}{2} \frac{4\cosh(\eta\lambda)\alpha(\eta) + \beta(\eta)}{4[\cosh(2\eta) - \cos(\eta\lambda)]^2}, \tag{86}$$

where [101], [102]

$$\begin{aligned}
\alpha(\eta) &= \sinh^2(\eta) - \cosh(\eta), \\
\beta(\eta) &= \cosh(\eta) + 2\cosh(2\eta) + 3\cosh(3\eta) - 2.
\end{aligned} \tag{87}$$

To be more specific, one can focus on two driving protocols discussed in the main text. We have numerically checked that for all these states and protocols the expression (83) is well defined and convergent.

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
