# Peer review of "Integrable Floquet dynamics"

_SciPost Physics, doi:SciPost Phys. 2, 021 (2017)_

## Round 2 · Referee Report · Anonymous (Referee 1) · 2017-2-26

Strengths

  1. Timely subject
  2. Novel point of view
  3. Attempt at classification

Weaknesses

  1. Ambiguous in scope
  2. No new concrete problems solved
  3. Unclear applicability

Report

The subject of Floquet systems is very timely and interesting. Integrability in this context is an interesting issue, which is not well studied. The authors present classes of periodically driven systems that satisfy the property that the effective Floquet Hamiltonian is integrable. I consider this paper interesting, and potentially suitable for publication in Scipost Physics. However, I would like the authors to consider the following points before publication:

  1. The title of the paper suggests a very general approach. However, the actual results are more restricted. Most of the paper (with the exception of the part on "Rotating frame") considers the two-step periodic process, for which the BCH expansion can be applied. I think the title should be chosen to more accurately reflect the content of the paper. The only element in the paper that conforms to the generality reflected in the title is the somewhat trivial definition of integrable Floquet system.

  2. The definition of integrable Floquet system used in the paper is very reasonable and intuitive. However, it has a strong limitation: the Floquet Hamiltonian is generally expected to be a very complicated nonlocal object even if the original Hamiltonian is local (e.g. a spin chain with short-range coupling). This seriously complicates identification of cases which are integrable since little is known about classification of nonlocal integrable Hamiltonians. The authors restrict themselves to the case of a two-step protocol and three very specific classes of algebras underlying the Hamiltonian, which do not have this problem.

  3. The only explicit problem considered in the paper is the Mathieu harmonic oscillator, which is nothing new, as the eventual solution is from the 1969 paper by Perelomov and Popov (ref. [43]). The authors do not present any other nontrivial example that could convince the reader about the usefulness of the approach. It is also unclear if there are real systems that fit into the author's framework; I think that NMR experiments can be considered for inspiration, as the two-step protocol is well-known there, but it remains to be seen whether the authors' approach can say anything nontrivial there.

  4. A more technical point is that the text has some grammatical problems and a number of typos, like "to a different Cartan subalgebras" towards the end of Section III.A, CMT instead of CTM after eqn. (31), FLoquet before eqn. (36), "discussed identified" in the first sentence of Section V, "irrespective if" before eqn. (63). At some points the text is not really comprehensible, for example at the end of Section III. B where it refers to an Appendix. The present paper has no appendices, so it must refer to an Appendix in either reference [39] or [41], but which one?

  5. References:

  6. when referencing GGE, [5] is an excellent work with respect to the role played by quasilocal charges, but in my opinion the original paper M. Rigol, V. Dunjko, V. Yurovsky, and M. Olshanii, Phys. Rev. Lett. 98, 050405 (2007) also deserves to be cited as the originator of the concept.
  7. fig. 2 is directly taken from the Perelomov-Popov paper, which must be acknowledged explicitly (just referencing [43] in the text when citing fig. 2 is not enough, at least in the usual practice of including artwork from other sources).
  8. when citing the XXZ quench action work ref. [89] of the Amsterdam group, the authors omit the parallel contribution by the Budapest group. The original PRL papers

B. Wouters, J. De Nardis, M. Brockmann, D. Fioretto, M. Rigol, and J.-S. Caux, Quenching the anisotropic Heisenberg chain: Exact solution and generalized Gibbs ensemble predictions, Phys. Rev. Lett. 113, 117202 (2014). B. Pozsgay, M. Mestyan, M. A. Werner, M. Kormos, G. Zarand, and G. Takacs, Correlations after quantum quenches in the XXZ spin chain: Failure of the generalized Gibbs ensemble, Phys. Rev. Lett. 113, 117203 (2014).

indeed appeared back-to-back; both groups contributed unique and important pieces to the full picture. The authors cite the long version of the Amsterdam group's paper as [89]; the corresponding paper by the Budapest group is

M. Mestyan, B. Pozsgay, G. Takacs and M.A. Werner, Quenching the XXZ spin chain: quench action approach versus generalized Gibbs ensemble, J Stat. Mech. 1504 (2015) P04001.

Requested changes

  1. The authors should give appropriate discussion of the issues raised in points 1-2 of the report, consider to change the title to reflect more the eventual scope of the results, and discuss the limitations of their approach more explicitly. In regard to point 3, the paper would be much improved by including at least one nontrivial and explicit novel example, or if the authors otherwise point out some more specific interesting systems where their approach can give new results.

  2. The authors should eliminate grammatical mistakes and typos (as much as possible), and clarify the text where appropriate.

  3. I suggest including the reference to the original GGE paper, and the authors must properly acknowledge the independent contributions of different groups in the context of the quench dynamics of the XXZ chain. It is also necessary to indicate explicitly the provenience of fig. 2 in its caption. The authors must also make sure they have the necessary permissions to reproduce it (this may be given under the terms the original journal is published, which may also specify the proper way of acknowledging the source of the figure).

---

## Round 2 · Referee Report · Anonymous (Referee 2) · 2017-2-27

Strengths

  1. Timely subject and interesting questions.
  2. Potentially extend the use of integrable models to driven systems.

Weaknesses

  1. Quite unclear definition of integrability in Floquet systems.
  2. It is unclear if there are applications to physical driven many-body systems.

Report

The authors deal with Floquet systems whose Floquet Hamiltonian is “integrable” and therefore the driven time evolution is not chaotic. This is a very interesting and promising field of research and the conclusions they find are definitely worth a publication. However the article as it is written now has many unclear parts which need to be addressed. Here I list the main questions that a reader may have.

Requested changes

1) First of all, it is not clear how the authors define integrability of the Floquet Hamiltonian. In the introduction they say they find “Floquet integrable systems in which one can define a local unfolded Floquet Hamiltonian”, but then below formula (2) they say “Integrability of HF in this paper will be understood as existence of enough conserved integrals of motion to be able to diagonalize it.” I personally struggle to see how these two definition are compatible. From a physical point of view they claim that Floquet integrable systems are those ones that do not heat up. Therefore I would expect that the eigenstates of the Floquet integrable Hamiltonian are not indistinguishable from a generic infinite temperature state, as it is usually the case for Floquet Hamiltonians. The authors do not mention this point (and I do not see how the fact that their Floquet Hamiltonians are diagonalizable is related to this) and therefore it is hard for me to understand what they mean for “they do not heat up”. 2) At page 3 the authors say “. Finally let us note that as with any other driven systems the physics can strongly depend on initial conditions, which can be also integrable or non-integarble”. Apart from the typo on the word integrable, it is not clear what do the authors mean with integrable initial conditions. 3) Figure 2 is taken from reference [43]. I am not sure if it is possible to use an already published figure without some consent. 4) Equation 33: what is \lambda? Operator B has no dependence on the spectral parameter. 5) Equation 36: The authors claim that this Hamiltonian is integrable. However, also due to the fact that their definition of integrability is unclear, I struggle to see the reason. For sure this Hamiltonian has not an extensive number of conserved operators, or at least if it does the authors should explain why. 6) Still on this equation: the operator B scales as L^2, with L the system size, while we expect the series \sum_n a_n Q_n to be convergent and therefore to be extensive in L. The authors should comment about this. Does this mean that the coupling constant b should scale as 1/L ? 7) Section IV.B is not well written. Many different notions and ideas are exposed without a clear picture of what is done. I suggest to organize this section in a clearer way and to expose clearly the main ideas/aims. 8) Formula 50: it should be a scalar product between the Pauli matrices. 9) Section IV.C The authors say “ However, the convergence of this formal expression should be checked for every state |Ψ0i separately. For this reasons we avoid presentation of these formal expressions”. I believe this should not prevent them to at least check their expression on a simple state, for example an infinite temperature state or a simple product state. That also would partially adress question 6). 10) Just below the authors notice that B has diverging matrix elements. Therefore a regularization has to be provided for B. This however seem strongly dependent on the boundary conditions. The authors should comment about this. 11) Just below eq 11: what is the matter of fact ? 12) Eq 60: It is not clear to me why the authors compute the rotating frame Hamiltonian and not directly the Floquet one. I kindly ask the authors to clarify a bit more their logic here in this section. 13) Non-numbered equation above eq 61: what is the variable n on the right hand side? 14) Eq 62) what is \bar{H}_{rot} ? Could the authors explain why in this case the Floquet Hamiltonian is the same as the rotation frame one? 15) The Floquet Hamiltonian 61 is time-independent. In the introduction the authors says “Using this criterion any Floquet system, which can be mapped to a static system via a local rotation (e.g. a static system in the rotating frame) is integrable because its folded spectrum contains infinitely many level crossings” therefore I would now expect that this Hamiltonian 61 falls in this class of integrable Hamiltonians. Could the authors comment about that? 16) Beginning of section V: I assume the authors meant “discussed and identified”.

Finally, as a general remark, I believe this paper contains many innovative ideas and interesting arguments, but it seems hard to find some physical application. Indeed it is not clear at all how the Floquet protocols introduced here are physically realizable (how to realize a time evolution with the Boost operator?). This is a general point where I invite the authors to comment.

  • validity: good
  • significance: ok
  • originality: good
  • clarity: low
  • formatting: acceptable
  • grammar: good

Author:  Vladimir Gritsev  on 2017-04-10  [id 114]

(in reply to Report 2 on 2017-02-27)
Category:
reply to objection

new version is available now: https://arxiv.org/abs/1701.05276v3

Attachment:

ref_reports.pdf

---

## Round 3 · Referee Report · Anonymous · 2017-4-24

Strengths
See previous report.
Weaknesses
See previous report.
Report
There is only one point where I am still not satisfied, which concerns the statement about the locality of the Floquet Hamiltonian. It is physically clear how the absence of heating is related to integrability, but I do not see why this has anything to do with locality. I still expect that the Floquet Hamiltonian is generally non-local and as far as I know this is revealed by explicit perturbation theory calculations.
However, this is not of much consequence to the new version of the paper, since it is made clear in the text that the classes of systems they present are examples rather than an exhaustive classification, and for these cases the Floquet Hamiltonian happens to be local and the integrals of motion can be found analytically.
Given that I am satisfied with the rest of their answers, I do recommend the publication of the revised version in Scipost Physics.
Requested changes
None.
Author: Vladimir Gritsev on 2017-05-16 [id 135]
(in reply to Report 2 on 2017-05-08)We thank our referee for careful reading, useful suggestions and interesting questions. We implemented his/her comments into a new version and added a paragraph (p.13, left column) as a reply to the first question. However, at this point we do not know if quasi-local conserved charges are important or not, so we did not speculate much on that issue.
Attachment:
ESF_III.pdf

---

## Round 3 · Referee Report · Anonymous · 2017-5-8

Strengths
Same as previous report
Weaknesses
--
Report
The authors have implemented the suggestions contained in the previous reports and I believe the paper is now almost ready for submission. However I still have some comments before publication. There are still few typos and therefore I recommend the authors to double check the manuscript again. For example:
-formula 71 should contain b0 and not lambda0
-Y \equiv = above eq 57
Moreover I believe that the abstract should be less technical and it should mention the possible experimental realizations of
the protocols they introduce in the manuscript (see the reports reply provided by the authors).
Finally a more general comment: In such “boost models” introduced by the authors there is a revival of the local conserved charges which implies that energy (and the other local charges) does not change between different periods and therefore there is no heating. However what can we say about the state (namely all correlation functions) a those times T_n? It is indeed now known that a Bethe state of a XXZ chain in the thermodynamic limit is not only fixed by the charges Q_n but also by the expectation values of all the quasi-local charges, (Ref [6] and J. Stat. Mech. (2016) 063101), so what can we say about these charges in these Floquet systems? Can we perhaps say that the state at time T_n is the maximal entropy state given the constrains of the local charges Q_n (as constructed in J. Stat. Mech. (2013) P07012 and Phys. Rev. Lett. 113, 117202 (2014)) ? I invite the author to provide a related discussion.
Requested changes
1)Correct typos like
-formula 71 should contain b0 and not lambda0
-Y \equiv = above eq 57
2)Rephrase the abstract
3)Add a discussion on the state at the times T_n during the time evolution given by the boost models.

---

## Editorial Decision

published